# Deep Active Learning with Manifold-Preserving Trajectory Sampling

## Abstract

Active learning (AL) is for optimizing the selection of unlabeled data for annotation (labeling), aiming to enhance model performance while minimizing labeling effort. The key question in AL is which unlabeled data should be selected for annotation. Existing deep AL methods arguably suffer from bias incurred by labeled data, which takes a much lower percentage than unlabeled data in AL context. We observe that such an issue is severe in different types of data, such as vision and non-vision data. To address this issue, we present Manifold-Preserving Trajectory Sampling (MPTS), which combines manifold regularization with trajectory sampling to address bias accumulation in active learning. By doing so, we expect to effectively correct the bias incurred by labeled data, which can cause a biased selection of unlabeled data. Despite its focus on manifold, the proposed method can be conveniently implemented by performing distribution mapping with MMD (Maximum Mean Discrepancies). Extensive experiments on various vision, non-vision and video benchmark datasets demonstrate the superiority of our method.

## 1 Introduction

Active learning (AL) has emerged as a powerful paradigm to mitigate data annotation costs across a wide range of machine learning applications, including medical diagnosis, autonomous driving, natural language processing, and computer vision. By enabling models to query the most informative unlabeled samples, AL can significantly improve performance with minimal supervision—an advantage particularly vital in scenarios where expert labeling is expensive, labor-intensive, or inherently limited.

Recent advances in deep active learning have focused primarily on uncertainty-based acquisition strategies. Techniques such as Monte Carlo (MC) dropout Gal & Ghahramani (2016), Bayesian neural networks (BNNs) Gal et al. (2017), and model ensembles Beluch et al. (2018); Czarnecki (2015) attempt to quantify epistemic uncertainty through stochastic forward passes, posterior modeling, or variance across multiple model instances. While effective to a degree, these methods suffer from two fundamental limitations that restrict their scalability and generalization.

The first issue stems from a subtle but critical distinction in active learning bias. While we intentionally want to bias our sample selection toward uncertain examples, we don't want this process to distort how our model understands the underlying data structure. When labeled data represent only a tiny fraction of the full dataset, the model's internal feature representations can drift away from what they would learn from the complete data distribution. This creates a compounding problem: biased features lead to poor uncertainty estimates, which in turn lead to even more biased sample selection. As a result, the model becomes increasingly biased toward a subspace that deviates from the true manifold, degrading both representation quality and uncertainty calibration. The second issue stems from reliance on restrictive assumptions or architecture-specific implementations—such as Gaussian posteriors in BNNs or the insertion of dropout layers in MC dropout—which hinder applicability across domains and model types.

To overcome these challenges, we propose **Manifold-Preserving Trajectory Sampling (MPTS)**, a novel active learning framework that simultaneously corrects sampling bias and enables robust uncertainty estimation without requiring architectural modifications or explicit distributional assumptions. MPTS introduces two key components: (1) a *manifold-preserving regularization* scheme that aligns feature representations of labeled and unlabeled data using Maximum Mean Discrepancy

(MMD), ensuring that the learned feature space remains faithful to the data manifold; and (2) a *trajectory-based parameter sampling* method that aggregates model checkpoints along optimization paths near local minima, capturing diverse posterior modes while avoiding the pitfalls of handcrafted uncertainty modeling.

We validate MPTS through extensive experiments across a broad range of visual, tabular, and video benchmarks. Results show that our method consistently outperforms state-of-the-art AL techniques, achieving higher accuracy with significantly fewer labeled samples. This highlights the importance of addressing sampling bias in uncertainty estimation and underscores the effectiveness of trajectory-based parameter exploration. We combine MMD regularization with trajectory sampling to address a specific problem in active learning: how bias accumulates over multiple rounds when models learn representations from increasingly skewed labeled datasets. While these techniques have been explored separately in other contexts, their integration provides an effective solution for maintaining feature quality throughout iterative active learning cycles.

**Our key contributions are threefold:**

- We introduce **MPTS**, the active learning framework that jointly integrates manifold-preserving regularization and trajectory-based parameter sampling. Unlike prior methods that rely on explicit posterior modeling or architectural constraints, MPTS provides a lightweight yet powerful solution for robust uncertainty estimation, free from distributional or structural assumptions.

- We systematically uncover and address the overlooked challenge of *sampling bias accumulation* across active learning cycles. By enforcing global manifold alignment between labeled and unlabeled data, MPTS effectively prevents representation drift and ensures that the model remains anchored to the true data distribution throughout the learning process.

- Through extensive experiments on a wide spectrum of benchmarks—including image classification, video analysis, and tabular data—we demonstrate that MPTS consistently achieves state-of-the-art performance with significantly fewer labeled samples, establishing a new standard for label efficiency and cross-domain generalizability in active learning.

## 2 RELATED WORK

### 2.1 ACTIVE LEARNING

Active learning is a pivot research area in machine learning, focused on optimizing data annotations to enhance model performance with fewer labeled samples. Most AL methods mainly consider uncertainty as a crucial criterion to intelligently sample data that improves model's generalization. Such methods prioritize data points with high prediction variance or near the decision boundary, employing techniques like MC-Dropout Gal & Ghahramani (2016), Query-by-Committee (QBC) Gorriz et al. (2017), and adversarial training Ducoffe & Precioso (2018) to address overconfident deep neural networks Tong & Koller (2001); Sinha et al. (2019). Li et al. Li et al. (2024b) propose a noise stability method that measures output deviation when model parameters are perturbed. However, these methods do not address the broader issue of distribution bias that accumulates across active learning cycles. Influence-based AL approaches select data points based on their estimated impact on model performance, using schemes like Learning Loss Yoo & Kweon (2019), and the Influence Function Koh & Liang (2017) that leverages gradient to estimate changes in prediction accuracy Liu et al. (2021); Wang et al. (2022). Besides, BADGE Ash et al. (2021) also aims to select uncertain data by evaluating gradient. Many deep AL methods resort to auxiliary models to estimate data uncertainty. Typical works include VAAL Sinha et al. (2019) which uses an auxiliary auto-encoder, and GCNAL Caramalau et al. (2021) that employs a graph network as the auxiliary model. Unlike these methods, the Coreset Sener & Savarese (2018) is free of any auxiliary models, but suffering from a slow optimization process (e.g., solving a classical K-center or 0-1 Knapsack problem) during data selection. Several other works, such as Zhang et al. (2020), rely on complicated training fashion (e.g., adversarial), and it will be challenging if using such methods on a different data format (e.g., 3D medical images of voxels) other than 2D natural images.

## 2.2 Posterior Approximation for Bayesian Neural Networks

Bayesian Neural Networks (BNNs) are designed to provide robust uncertainty estimates by treating the network's parameters as probabilistic distributions rather than fixed values. This approach is essential for capturing uncertainty in tasks like active learning. Several works Maddox et al. (2018; 2019); Lindén et al. (2020) propose to estimate posterior distributions by averaging the training checkpoints. To this end, they use Stochastic Weight Averaging (SWA) Demir et al. (2024) to perform the averaging operation, improving the uncertainty estimation. SSVI Li et al. (2024a) maintains sparse Bayesian models throughout training and inference phases. This approach reduces computational costs. These methods offer practical solutions for reliable uncertainty estimation in deep networks. Notably, our method has a very low level of similarity with these methods, as we propose a brand new solution to estimate the posterior considering both labeled and unlabeled data simultaneously.

## 3 Method

To address the fundamental challenges of data bias accumulation and distributional assumption dependence in multi-cycle active learning, we propose MPTS (Manifold-Preserving Trajectory Sampling), a novel framework that ensures unbiased posterior estimation while providing assumption-free parameter sampling.

### 3.1 Problem Analysis

**Active Learning Bias Accumulation.** We formalize the multi-cycle active learning problem as follows. Given an initial labeled set $\mathcal{L}$ and a large unlabeled pool $\mathcal{U}$, the objective is to iteratively select the most informative subset $X_N \subset \mathcal{U}$ for annotation. After expert annotation, the labeled set is augmented as $\mathcal{L} \leftarrow \mathcal{L} \cup X_N, Y_N$ where $Y_N$ denotes the corresponding labels. For uncertainty-based sample selection, we estimate prediction uncertainty using entropy:

$$H(x) = -\sum_{c=1}^{C} p(y = c|x, \theta) \log p(y = c|x, \theta) \tag{1}$$

However, reliable uncertainty estimation requires accurate posterior distributions. Bayesian neural networks incorporate parameter posterior distributions:

$$p(y = c|x, D) = \int p(y = c|x, \theta) p(\theta|D) d\theta \tag{2}$$

The critical issue lies in posterior estimation. Applying Bayes' rule:

$$p(\theta|D) = \frac{p(\theta)p(D|\theta)}{p(D)} \propto p(\theta)p(D|\theta) \tag{3}$$

For discriminative models with dependency chain $X \to Z \to Y$, the likelihood factorizes as:

$$p(D|\theta) = p(X|\theta)p(Z|X, \theta)p(Y|X, Z, \theta) \tag{4}$$

The core issue becomes clear when we examine what happens during training. Most active learning methods focus entirely on the classification layer - they estimate $p(Y|X, Z, \theta)$ using only labeled samples. But they largely ignore whether the feature extractor $p(Z|X, \theta)$. learned from these labeled samples actually captures meaningful patterns from the full dataset. Over multiple rounds, the model becomes increasingly specialized at representing only the types of examples it has seen, making it progressively worse at understanding the broader data landscape. This doesn't mean we should abandon uncertainty-based selection - it means we need to prevent the model's feature understanding from becoming too narrow.

### 3.2 Manifold-Preserving Regularization

To address both the feature distribution bias and the rigid distributional assumptions, our MPTS framework consists of two complementary strategies: Strategy 1: Manifold-Preserving Regularization corrects the biased feature distribution $p(Z|X, \theta)$ by leveraging abundant unlabeled data.

Strategy 2: Trajectory-Based Parameter Sampling captures diverse posterior modes without explicit distributional assumptions.

A natural question arises: won't correcting this bias just push us back toward random sampling? The answer lies in understanding what we're actually correcting. We're not trying to make our selected samples look like random samples - we still want to pick uncertain, informative examples. Instead, we're ensuring that our model's internal understanding of data patterns stays grounded in the full dataset reality, even when it only learns labels from a selective subset. Since existing methods only use biased labeled data $\mathcal{L}$ to learn feature representations, we need to regularize the feature distribution to align with the true data manifold.

We enforce the feature distribution learned from labeled data to approximate that from the complete dataset:

$$Z_{\mathcal{L}} = f_e(X_{\mathcal{L}}; \theta) \approx Z_* = f_e(X_{\mathcal{L} \cup \mathcal{U}}; \theta) \tag{5}$$

where $f_e$ denotes the feature extractor component of the network.

We employ Maximum Mean Discrepancy (MMD) to measure and minimize this distribution gap:

$$\text{MMD}(Z_{\mathcal{L}}, Z_*) = \sup_{h \in \mathcal{H}} \{\mathbb{E}_{z \sim Z_{\mathcal{L}}}[h(z)] - \mathbb{E}_{z \sim Z_*}[h(z)]\} \tag{6}$$

The training objective combines supervised learning with manifold preservation:

$$\mathcal{L}_{total} = \mathcal{L}_{ce}(X_{\mathcal{L}}, Y_{\mathcal{L}}) + \lambda \text{MMD}^2(Z_{\mathcal{L}}, Z_*) \tag{7}$$

The effectiveness of MMD regularization in correcting active learning bias stems from its ability to enforce distributional alignment in reproducing kernel Hilbert spaces (RKHS). Let $\mu_{\mathcal{L}}$ and $\mu_*$ denote the feature distributions induced by labeled and complete datasets respectively. The MMD distance provides an unbiased estimator of the distributional discrepancy:

$$\text{MMD}(Z_{\mathcal{L}}, Z) = \sup_{h \in \mathcal{H}} (\mathbb{E}_{z \sim Z_{\mathcal{L}}}[h(z)] - \mathbb{E}_{z \sim Z}[h(z)]) \tag{8}$$

For universal kernels, $MMD = 0$ if and only if the two distributions are identical. In the context of active learning, the biased sampling process creates a distribution shift: $\mu_{\mathcal{L}} \neq \mu_*$. By minimizing $\text{MMD}^2(Z_{\mathcal{L}}, Z_*)$, we enforce the learned feature extractor to satisfy:

$$\lim_{|\mathcal{L}| \to \infty} \text{MMD}(\mu_{\mathcal{L}}, \mu_*) = 0 \tag{9}$$

This constraint prevents the feature representation from overfitting to the biased labeled subset and maintains consistency with the underlying data manifold. The training objective combines supervised learning with manifold preservation:

$$\mathcal{L}_{total} = \mathcal{L}ce(X_{\mathcal{L}}, Y_{\mathcal{L}}) + \lambda \text{MMD}^2(Z_{\mathcal{L}}, Z_*) \tag{10}$$

Convergence Analysis. We analyze the convergence properties of our manifold-preserving objective. Under standard smoothness assumptions on the loss function and bounded feature spaces, the MMD regularization term is Lipschitz continuous with respect to network parameters. Specifically, for a fixed kernel k and bounded feature domain $\mathcal{Z}$, there exists a constant $L > 0$ such that:

$$\left| \text{MMD}^2(Z_{\mathcal{L}}(\theta_1), Z(\theta_1)) - \text{MMD}^2(Z_{\mathcal{L}}(\theta_2), Z(\theta_2)) \right| \leq L |\theta_1 - \theta_2| \tag{11}$$

To empirically validate our convergence analysis, Figure 1 demonstrates the MMD convergence behavior across different regularization strengths. The results confirm the theoretical trade-off established: smaller $\lambda$ values achieve rapid convergence within 30 epochs but stabilize at suboptimal MMD values around 0.8, indicating insufficient distributional alignment. Conversely, larger $\lambda$ values exhibit extended convergence periods lasting beyond 60 epochs while ultimately reaching superior alignment with MMD values.

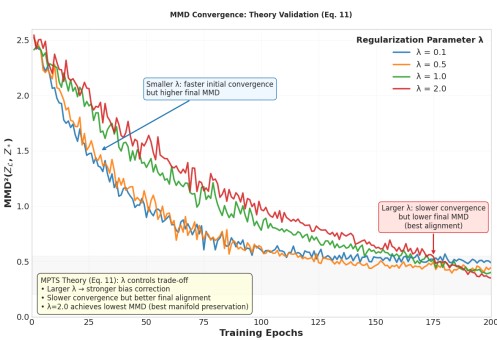
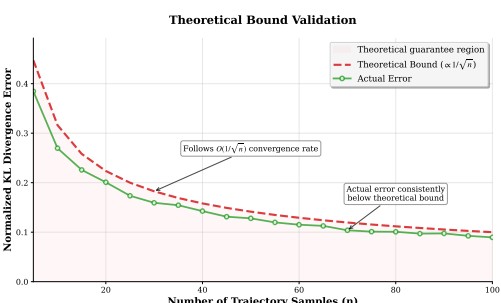

Figure 1: MMD convergence under different regularization strengths. Higher $\lambda$ values exhibit slower initial convergence but achieve superior final alignment between learned and target distributions. The trade-off demonstrates that stronger regularization ($\lambda$=2.0) yields the lowest asymptotic MMD despite delayed convergence dynamics.

Figure 2: Validation of the theoretical bound in Equation 12. The actual approximation error (solid line) follows the predicted $O(n^{-1/2})$ convergence rate and stays within the theoretical guarantee (dashed line), confirming our analysis.

### 3.3 TRAJECTORY-BASED PARAMETER SAMPLING

Traditional methods assume fixed posterior forms, limiting their flexibility. We collect parameters during optimization to capture multiple posterior modes. Neural networks find various good solutions during training, each representing a valid data hypothesis. We modify Stochastic Weight Averaging (SWA) for active learning through two phases. First, we train the network to convergence with standard optimization. Second, we apply a cyclic learning rate that alternates between high and low values. This cycling explores different parameter regions near the converged solution. We collect parameter snapshots at each epoch's end rather than at random intervals. Since training data stays fixed within epochs, this timing captures real model variations instead of data shuffling noise. New labeled samples are added only between epochs, preventing interference across SWA collection phases. During exploration, we save parameter checkpoints $\theta_{t_i}{}_{i=1}^{n}$ from epoch endpoints. Each checkpoint represents a local minimum state. We typically collect 10-20 checkpoints based on dataset size. The cyclic schedule runs for about 25% of initial training epochs, balancing exploration with computational cost. This epoch-based approach reduces parameter variability within cycles while maintaining diversity across training phases. The collected parameter snapshots form a trajectory ensemble that theoretically approximates the posterior distribution. The trajectory ensemble approximates the true posterior with bounded error:

$$\mathbb{E}[\mathrm{KL}(p_{true}(y|x)|p_{ensemble}(y|x))] \leq \frac{C}{\sqrt{n}} + O(\epsilon_{opt}) \tag{12}$$

where $n$ is the number of trajectory samples and $\epsilon_{opt}$ is the optimization error.

The final prediction probability is computed by averaging predictions across all sampled parameters:

$$p(y = c|x, D) = \frac{1}{n} \sum_{i=1}^{n} p(y = c|x, \theta_{t_i}) \tag{13}$$

where each $\theta_{t_i}$ is sampled from the trajectory optimization. This ensemble provides a better estimate of the true predictive distribution than any single model, with prediction variance naturally capturing epistemic uncertainty.

When combining manifold-preserving regularization with trajectory sampling, our framework inherits favorable properties from both components. The MMD regularization ensures consistency with the true data distribution, while trajectory sampling captures parameter uncertainty. This dual mechanism maintains reliable uncertainty estimates throughout active learning cycles, even as biased samples accumulate in the labeled set.

**Theoretical Validation.** To verify our analysis, we examine whether the trajectory ensemble exhibits the convergence behavior predicted in Equation 12. Figure 2 shows the approximation error decreases with trajectory sample size n. The results confirm our theoretical prediction: the actual error follows the expected $O(1/\sqrt{n})$ rate and remains below the theoretical bound. This validates that our bound provides meaningful guarantees rather than loose estimates.

**Computational Efficiency.** Our integrated approach achieves bias correction and uncertainty quantification with manageable overhead: Training Complexity: $O(E \cdot |\mathcal{L}| \cdot d + E \cdot b^2 \cdot k)$ per active learning round, where the MMD term adds only $O(b^2 \cdot k)$ overhead since $b^2 \cdot k \ll |\mathcal{L}| \cdot d$. Storage Requirements: $O(n \cdot d + |\mathcal{U}| \cdot k)$ for trajectory parameters and feature caching. Efficiency Advantage: Compared to ensemble methods requiring $O(m \cdot E \cdot |\mathcal{L}| \cdot d)$ with $m$ separate models, our single-model approach with $n$ trajectory samples achieves comparable uncertainty quality at $O(n/m)$ relative cost, making it practically viable for large-scale active learning scenarios.

Table 1: A summary of various AL settings we use in the experiments.

| Dataset | Pool Size | Label Size | Input | Initial Instances | Budget | Backbone | Initialization |
|---------|-----------|------------|-------|-------------------|--------|----------|----------------|
| CIFAR10 | 50,000 | 10 | $32 \times 32$ | 100 | 100 | ResNet-18 | Random |
| MNIST | 50,000 | 10 | $28 \times 28$ | 100 | 100 | MLP | Random |
| SVHN | 50,000 | 10 | $32 \times 32$ | 100 | 100 | ResNet-18 | Random |
| Mini-ImageNet | 48,000 | 100 | $84 \times 84$ | 1000 | 1000 | ViT-Small | Pre-trained |
| OpenML-6 | 18,000 | 26 | 16 | 100 | 100 | MLP | Random |
| OpenML-155 | 50,000 | 9 | 10 | 100 | 100 | MLP | Random |
| HMDB | 5310 | 102 | 32 | 204 | 204 | MViT | Pre-trained |

## 4 EXPERIMENTS

Here we introduce a series of experiments conducted to validate the proposed method. To make the evaluation more comprehensive, we consider multiple datasets, backbone models, as well as different AL settings. We refer readers to Table 1 for details.

### 4.1 DATASETS AND BASELINES

**Datasets.** As most deep AL methods have been evaluated in computer vision challenges, we also adopt four widely used benchmark vision datasets to evaluate our method, including MNIST Lecun et al. (1998), CIFAR10 Krizhevsky et al. (2009), SVHN Netzer et al. (2011), and Mini-ImageNet Ravi & Larochelle (2016). In addition, we also incorporate two typical non-vision datasets for the evaluation, namely OpenML-6 OpenML (2021) and OpenML-155 OpenML (2021), which are tabular datasets from the OpenML repository, including structured data with mixed types of features. To verify the generalization of our method, we also evaluated the video dataset HMDB Kuehne et al. (2011).

**Baselines.** The baseline methods that we consider in this paper can be categorized into three groups. The first group resorts to estimate data uncertainty based on posterior, including Entropy Wang & Shang (2014), BALD Gal et al. (2017), BADGE Ash et al. (2020). The second group designs customized methods to evaluate data uncertainty, including Coreset Sener & Savarese (2018), CDAL Agarwal et al. (2020), and Feature Mixing Parvaneh et al. (2022). The third group relies on auxiliary models and/or special training fashion (e.g., adversarial), including Adversarial Deep Fool Ducoffe & Precioso (2018) and GCNAL Caramalau et al. (2021). In addition, Random selection is also included as it is a straightforward yet effective method in several scenarios.

**Models.** We use three types of deep models as the backbones. Specifically, we use MLP Ash et al. (2020) for MNIST, ResNet-18 He et al. (2016) as a typical CNN for CIFAR10 and SVHN, and vision transformer (ViT) Alexey (2020) as a typical foundation model for Mini-ImageNet. We also use MLP for the two non-vision datasets. We use MViT Fan et al. (2021) to the video dataset.

### 4.2 EXPERIMENTAL SETTINGS

For each dataset, following the common practice in AL literature, we randomly select a small portion of data as initial samples and annotate them. The number of such samples is 100 for all the datasets,

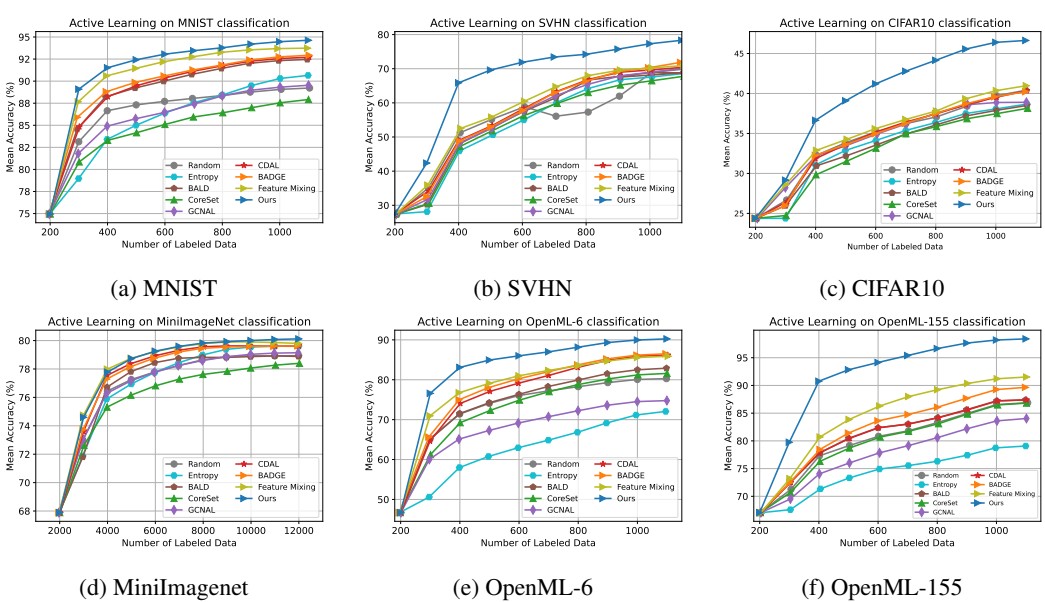

Figure 3: This is the main caption for all six sub-figures.

Table 2: Ablation study of Class-wise vision datasets.

| Class (# of samples) | 0 (980) | 1 (1135) | 2 (1032) | 3 (1010) | 4 (982) | 5 (892) | 6 (958) | 7 (1028) | 8 (974) | 9 (1009) |
|---|---|---|---|---|---|---|---|---|---|---|
| MNIST | 97.44 | 97.14 | 90.69 | 91.98 | 94.50 | 93.94 | 92.48 | 95.91 | 91.78 | 87.90 |

| Class (# of samples) | 0 (1744) | 1 (5099) | 2 (4149) | 3 (2882) | 4 (2523) | 5 (2384) | 6 (1977) | 7 (2019) | 8 (1660) | 9 (1595) |
|---|---|---|---|---|---|---|---|---|---|---|
| SVHN | 70.35 | 80.15 | 85.32 | 80.11 | 79.62 | 66.82 | 70.10 | 63.55 | 86.42 | 71.72 |

except the Mini-ImageNet in which we use 1000 initial samples. Then in each AL round (covering both model training and data selection phases), we select 100 unlabeled samples (labeling budget) for all the datasets, except the Mini-ImageNet where we select 1000 unlabeled samples for initial annotation. When MLP or CNN is used as the backbone, within each AL round, we train the model for 100 epochs. When pre-trained ViT is used, we fine-tune it for 1000 epochs within each AL round. We adopt a learning rate of $1e-3$ for vision datasets and $1e-4$ for non-vision datasets. The batch size is set to 64 for all the experiments. Notably, we train the MLP and CNN from scratch, whereas we fine-tune the pre-trained ViT following the practice in Parvaneh et al. (2022) for a fair comparison. To reduce randomness, we repeat each experiment for 5 times and average the results as the final one. We observe that some baseline methods occasionally perform worse than random selection, particularly in early rounds with very limited labeled data. This phenomenon, documented in prior active learning literature, occurs when uncertainty estimates become unreliable due to insufficient training data. To ensure fair comparison, we report results starting from the second active learning round (after 200 labeled samples) when all methods have stabilized.

## 4.3 RESULTS AND ANALYSIS

We first examine how our method preserves data manifold structure during active learning selection. Figure 4 shows t-SNE visualizations of MNIST data before and after applying our selection strategy. The left panel displays the complete dataset (2000 samples) with natural clustering of digit classes, while the right panel shows our selected subset.

Figure 3 demonstrates that MPTS consistently outperforms baseline methods across all evaluated datasets. The method adapts well to different architectures, working effectively with MLP on MNIST, ResNet-18 on CIFAR10 and SVHN, and ViT on Mini-ImageNet. This consistent performance across diverse model types provides initial evidence for the method's broad applicability.

Building on it, we observe that the performance gains correlate with both budget constraints and data complexity. Under limited annotation budgets (100 samples per round), MPTS maintains clear

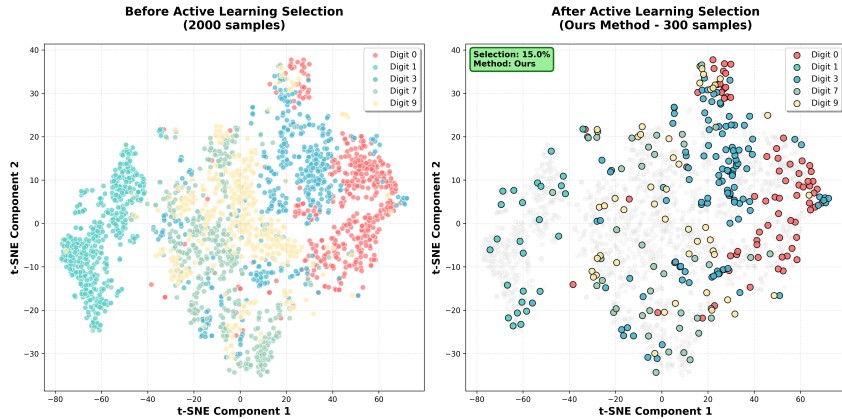

Figure 4: t-SNE visualization demonstrating manifold preservation. Our method selects 15% of MNIST samples (right) while maintaining the distributional structure of the complete dataset (left), with representative coverage across all digit classes and decision boundaries.

advantages on MNIST, CIFAR10, and SVHN, which has direct relevance for cost-sensitive applications like medical imaging. More importantly, the improvement margins vary systematically with dataset complexity. While MNIST shows modest gains due to its relative simplicity, CIFAR10 and SVHN exhibit larger performance differences. This pattern suggests that manifold preservation becomes increasingly valuable as visual complexity grows. Even when scaling to larger budgets (1000 samples on Mini-ImageNet), MPTS retains its competitive edge, addressing a common limitation where active learning methods lose effectiveness at higher annotation volumes.

The benefits extend beyond computer vision to other data modalities. On tabular datasets (OpenML-6 and OpenML-155), MPTS achieves substantial improvements over baselines. Since these experiments use MLP architectures, the results 3 reveal that simpler models are particularly vulnerable to distribution bias, which our regularization strategy effectively addresses. This finding connects back to the MNIST results, where the combination of simple data and simple models limits the observable benefits. To further validate cross-domain effectiveness, we evaluated MPTS on video classification using the HMDB Kuehne et al. (2011) dataset with MViT Carreira & Zisserman (2017) backbone. Table 3 shows consistent improvements over ALFA-Mix, a specialized video active learning method, with performance gaps widening from 1.15% (204 samples) to 2.47% (1530 samples).

The comparative analysis reveals key methodological advantages. Unlike GCNAL, which requires auxiliary graph networks alongside the main model, MPTS achieves better results using only the base architecture. Table 4 compares our approach with ASWA, showing that methods focusing solely on weight averaging without considering manifold structure achieve lower performance.

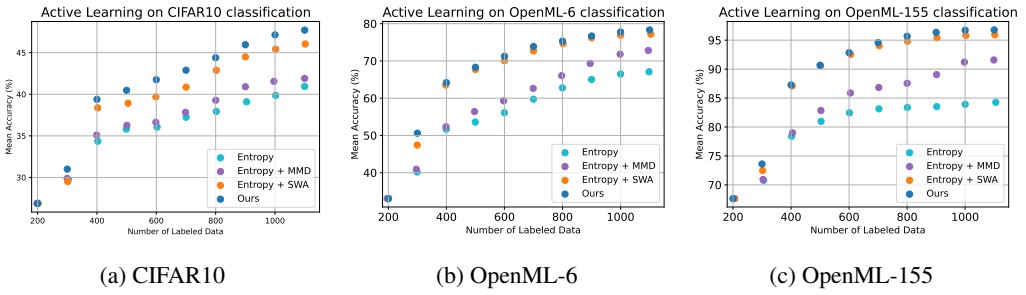

| (a) CIFAR10 | (b) OpenML-6 | (c) OpenML-155 |

Figure 5: Ablation study of the methods on vision and non-vision datasets

## 4.4 ABLATION STUDY

We evaluate each component's contribution by progressively removing manifold-preserving regularization and trajectory sampling. Figure 5 shows the results across three datasets. Using entropy-based selection alone produces limited improvements and unstable performance. The method strug-

Table 3: Compare the accuracy of our method and ALFA-Mix.

| Method | Dataset | AL Rounds | | | |
|---|---|---|---|---|---|
| | | 204 | 408 | 765 | 1530 |
| ALFA-Mix | HMDB | 61.45 | 67.64 | 73.31 | 76.35 |
| **Ours** | | **62.60** | **68.95** | **74.96** | **78.82** |

Table 4: Compare our method with ASWA on the CIFAR10 and OpenML6 datasets.

| Method | Dataset | AL Rounds | | | |
|---|---|---|---|---|---|
| | | 400 | 600 | 800 | 1000 |
| ASWA | CIFAR10 | 25.67 | 36.99 | 39.67 | 43.75 |
| **Ours** | | **38.45** | **41.15** | **42.98** | **45.82** |
| ASWA | OpenML6 | 57.35 | 66.96 | 70.15 | 73.20 |
| **Ours** | | **58.74** | **67.22** | **74.07** | **76.50** |

Table 5: Gap (%) between training and test accuracy after each AL cycle, evaluated on CIFAR10.

| Budget (%) | 20 | 30 | 40 | 50 | 60 |
|---|---|---|---|---|---|
| Random | 6.54 | 7.68 | 9.95 | 10.36 | **10.10** |
| Entropy | 6.61 | 8.63 | 10.52 | 12.51 | 15.19 |
| BALD | **6.12** | 7.30 | 8.98 | 10.32 | 11.42 |
| CoreSet | 6.39 | 7.23 | 9.68 | 11.45 | 12.54 |
| Badge | 6.53 | 8.65 | 10.24 | 12.39 | 14.00 |
| CDAL | 8.00 | 8.66 | 10.82 | 15.27 | 16.40 |
| **Ours** | 6.23 | **7.20** | **8.95** | **10.22** | 10.85 |

Table 6: Ablation study of Class-wise non-vision datasets.

| Class (# of samples) | 0 (71) | 1 (75) | 2 (81) | 3 (68) | 4 (75) | 5 (74) | 6 (76) | 7 (61) | 8 (72) |
|---|---|---|---|---|---|---|---|---|---|
| OpenML-6 | 92.95 | 77.46 | 80.00 | 90.12 | 88.23 | 88.00 | 78.37 | 82.89 | 72.13 |

| Class (# of samples) | 9 (72) | 10 (75) | 11 (65) | 12 (69) | 13 (64) | 14 (68) | 15 (77) | 16 (94) | 17 (88) |
|---|---|---|---|---|---|---|---|---|---|
| OpenML-6 | 95.83 | 86.11 | 73.84 | 84.05 | 87.50 | 77.94 | 92.20 | 70.21 | 85.22 |

| Class (# of samples) | 18 (74) | 19 (91) | 20 (93) | 21 (89) | 22 (71) | 23 (101) | 24 (78) | 25 (96) | 26 (57) |
|---|---|---|---|---|---|---|---|---|---|
| OpenML-6 | 83.78 | 81.31 | 89.24 | 94.38 | 87.32 | 95.04 | 91.02 | 87.50 | 84.21 |

gles with overfitting on certain datasets, indicating that pointwise uncertainty measures cannot capture underlying distribution structures. Adding MMD regularization (Entropy+MMD) significantly improves stability and consistency. This demonstrates that aligning feature distributions between labeled and unlabeled data effectively reduces selection bias. Incorporating trajectory sampling (Entropy+SWA) also improves over the baseline, but less than MMD regularization. Tables 2 and 6 provide class-wise analysis on vision and non-vision datasets. Our method maintains consistent performance across different classes, including those with fewer samples or complex structures.

## 4.5 GENERALIZATION ANALYSIS

We examine model generalization by measuring the training-test accuracy gap across active learning cycles. Early cycles with limited data often produce unstable gap measurements due to noise and random factors. To obtain reliable estimates, we scale up the CIFAR10 experiment with $50\times$ larger initial sets and $25\times$ larger selection budgets. Table 5 shows the accuracy gaps at different annotation percentages. Baseline methods exhibit increasing gaps as budgets grow, indicating progressive overfitting. For example, BALD maintains a small gap (6.12%) at 20% budget but degrades at higher budgets (11.42% at 60%). This pattern reflects the accumulation of selection bias over multiple cycles.

## 5 CONCLUSION

We identify a risk in deep active learning where uncertainty estimation relies solely on biased labeled data, leading to progressive deviation from the true data manifold across learning cycles. To address this challenge, we introduce MPTS, a framework that corrects feature distribution bias through manifold-preserving regularization while capturing diverse posterior modes via trajectory-based parameter sampling. The manifold-preserving component is employed to align feature distributions between labeled and unlabeled data. Comprehensive experiments across vision datasets, tabular data, and video sequences confirm the effectiveness of our approach. Ablation studies validate that both manifold preservation and trajectory sampling contribute to performance improvements.

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
