# OpenReview forum: "Deep Active Learning with Manifold-Preserving Trajectory Sampling"
_ICLR.cc/2026/Conference — ICLR 2026 Conference Withdrawn Submission_

### Official Review · Reviewer_5Kb3 · 2025-10-21

**Soundness:** 2
**Presentation:** 2
**Contribution:** 2
**Rating:** 2
**Confidence:** 4

**Summary:**

Overview

The paper describes the use of two methodologies: manifold preserving regularization and trajectory-based parameter sampling for improving deep active learning. At the basis of this construction is the claim that standard uncertainty sampling criterion do not map the distribution well enough and introduce a bias. The authors also indicate architecture dependance as hindering performance.
The authors provide experimental results with standard up to 50K data sets. They also use several standard uncertainty sampling methods to compare their method with.
I have a disagreement with the authors on the problem statement and motivation, and I feel that their experimental results are not satisfactory, as their initial labeled sets form quite a large percentage of the overall data set and as such already map the distribution quite well. The paper is missing a fundamental demonstration of the problem, and its solution, even on a simple toy example to convince the readers. Graph based active learning is a missing baseline that follows the data manifold as well. Some missing formulation and definitions, also makes the flow difficult. Hence to this end I cannot recommend this manuscript for publication.

**Strengths:**

Strengths

Method supported by theoretical foundations in Bayesian inference
Empirical results show advantage of the method over selected baselines
The trajectory-based sampling seems like a novel method.

**Weaknesses:**

Weaknesses

1.	Active learning aims at optimizing the tradeoff between labeling effort and model accuracy. However, if we sample in a certain class distribution location that is not close to the decision boundary, I claim that this is a redundant sampling. The authors criterion to map distributions may result in redundant sampling, in my view. I would have much preferred it if the authors would introduce the well-known sampling tradeoff of exploration-exploitation in which the context of their work may fit in better. Mapping the distribution amounts to exploration, which allows us to discover new decision boundaries. But it is clear that at some point better gains in performance are obtained when refining the discovered decision boundaries and not in mapping the distribution, and this is where I find the authors consistent use of manifold regularization as problematic.

2.	It is also not entirely clear how manifold regularization is translated to an actual sampling criterion. There is no pseudo-code or even a sampling criterion stated in this paper.

3.	I think the paper can benefit from a toy example and demonstration of the problem and they solution (e.g. binary checkerboard example) Fig 4 doesn’t show clear class separation and in fact shows how decision boundaries are left uncovered.

4.	Related work and experimental baselines don’t cover graph-based active learning methods some of which address -well manifold and distribution, including in deep active learning. Please check:

a.	Graph-based Active Learning for Semi-supervised Classification of SAR Data, Kevin Miller and John Mauro and Jason Setiadi and Xoaquin Baca and Zhan Shi and Jeff Calder and Andrea L. Bertozzi

b.	Diffusion-based Deep Active Learning, Dan Kushnir, Luca Venturi

c.	Active-transductive learning with label-adapted kernels, D Kushnir

d.	Accelerated Deep Active Learning with Graph-based Sub- Sampling, Dan Kushnir and Shiyun Xu

5.	Related work didn’t cover the subject of trajectory sampling.

6.	There are several issues and questions on the experimental evaluations:

a.	The results cover quite a narrow set of small data sets, why not try ImageNet? Is the method limited in its applicability to large data sets?

b.	Table 1: The initial labeled sets and actively selected batches are quite big and Im not sure they actually represent a bias, unless the authors would like to show us how they are distributed among the classes. In particular, are the initial set of 100 and 1000 are equally distributed among the classes?

c.	There are no error bars reported or standard deviation in tables 3-6. It is standard practice.

d.	The gaps comparison in Table 5 don’t show a large advantage in removing bias for the suggested method when compared to other methods, also budget  is very high, so obviously there is similarity between the distributions. Im not fully convinced.

e.	Im not sure what Table 6 is trying to tell us.

7.	Missing definitions makes the paper hard to read:

a.	Eq (2) D is not defined,  \theta

b.	Eq (12): how is derived, or where from?

c.	Computational Efficiency: what are E, d \L b k? please provide definitions of variables you are using.

**Questions:**

please see above

---

### Official Review · Reviewer_uYmt · 2025-10-28

**Soundness:** 3
**Presentation:** 3
**Contribution:** 3
**Rating:** 6
**Confidence:** 5

**Summary:**

This paper reveals that existing AL methods overlook the fact that internal feature representations can drift away from what they would learn from the complete data distribution. Considering this, the authors make their contributions by proposing a manifold-preserving regularization scheme that aligns feature representations of labeled and unlabeled data using MMD.

**Strengths:**

1. The discussion of the issue raised by existing AL methods is motivated.
2. The proposed method is model-agnostic and can be easily integrated into existing AL frameworks.

**Weaknesses:**

1. The claim in Line 169-170, "Since existing methods only use biased labeled data L to learn feature representations" is not rigorous, as many existing AL methods also adopt unlabeled samples for learning feature representations.
2. The intuition behind the choice of MMD distance is missing.
3. It is unclear how to calculate Equation 8. More details are necessary.
4. The proposed method seems to be incremental, as MMD for manifold learning and stochastic Weight
Averaging (SWA) for active learning can be found in existing works.

**Questions:**

My main question is about why choose MMD as the distribution distance measurement? How about other solutions, e.g., KL distance and Wasserstein Distance？

---

### Official Review · Reviewer_sr7g · 2025-11-01

**Soundness:** 2
**Presentation:** 2
**Contribution:** 2
**Rating:** 4
**Confidence:** 5

**Summary:**

The paper proposes an active learning approach that integrates manifold preservation and a trajectory-based posterior estimation to address the selection bias issue in uncertainty-based active learning strategies. The paper identifies the problem of evaluating the uncertainty by focusing entirely on the classification layer, and uses manifold-preserving regularization to maintain the landscape of the feature representation. Instead of explicit posterior estimation, the paper uses a cyclic learning rate trajectory and sample parameters from a trajectory ensemble.

**Strengths:**

Although the selection bias in active learning is not a new concept, the paper provides a fresh angle with the analysis in section 3.1. The manifold-preserving regularization aligns well with the analysis. The experiments are conducted on multiple datasets with a variety of feature sizes and model backbones as well. The proposed method shows a good advantage in the given settings.

**Weaknesses:**

1. The theoretical analysis is not clear enough. Equation (9) is too general and not informative. Equation (11) uses strong assumptions without much specification. Equation (12) is only verified empirically. The second strategy, the trajectory-based parameter sampling, is also a largely heuristic approach and weakens the connection between the theoretical motivation and proposed method.

2. There is too little analysis on the active sampling process. While the manifold perspective is somewhat novel, many existing AL strategies balance informativeness and representativeness. It is unclear how the proposed method is better in these aspects.

3. The evaluation part also lacks comparison with informativeness-representativeness balancing approaches, such as WAAL, VAAL (Shui, Changjian, et al. "Deep active learning: Unified and principled method for query and training." International conference on artificial intelligence and statistics. PMLR, 2020., Sinha, Samarth, Sayna Ebrahimi, and Trevor Darrell. "Variational adversarial active learning." Proceedings of the IEEE/CVF international  conference on computer vision. 2019.) and other more recent AL baselines (the backbone choice is also too weak to compare with some of them, having sub-50% performance on CIFAR10).

**Questions:**

1. How does the motivation clearly connect to the proposed method?

2. How does the manifold perspective compare to informativeness-representativeness balancing approaches?

3. Is there a more detailed analysis of the sampling process?

---

### Official Review · Reviewer_Fw8n · 2025-11-02

**Soundness:** 3
**Presentation:** 3
**Contribution:** 3
**Rating:** 4
**Confidence:** 4

**Summary:**

The author proposed Manifold-Preserving Trajectory Sampling (MPTS) as a new active learning query selection strategy to effectively prevent biased selection of unlabeled data. In their framework, they combine maximum mean discrepancy regularization with trajectory-based parameter sampling to ensure a robust and unbiased uncertainty estimation. They verify the effectiveness of MPTS on image, video and tabular datasets.

**Strengths:**

1. Trajactory-based sampling ensures sampling diversity and is faster than ensembled methods.
2. The model outperform other baselines on lots of tasks, such as SVHN, CIFAR10 and two tabular datasets --OpenML-6 and OpenML-155.

**Weaknesses:**

1. The trade-off between computational cost and model performance gain wasn't well established. Eventhough the author provided their complexity analysis in section 3.3, it would be more direct if we can see the wall-clock time of sampling time for MPTS and other benchmarks.
2. The advantage of using MPTS becomes minor when it comes to pretraining tasks. And the current experiments scope are limited to toy image/tabular/video data. I'm not sure about its impact in more realistic scenarios.
3. The writing needs improvement. For example, ASWA is used as a benchmark but I cannot find its definition in the main text.

**Questions:**

1. I notice the performance gain brought by MPTS is not much in pretrained tasks such as Mini-ImageNet and HMDB. Can you share some insights?
2. what is ASWA? I didn't find the citation or full name of that method.
3. In figure 3c, the best CIFAR10's accuracy achieved is ~47%. Have you tried a larger budget or more rounds? Does MPTS keep leading with more budget?

---

### Note · Authors · 2025-11-17

I have read and agree with the venue's withdrawal policy on behalf of myself and my co-authors.